# The Integration of AI into the Nursing Process: A Comparative Analysis of NANDA, NOC, and NIC-Based Care Plans

**DOI:** 10.3390/nursrep15060186

**Published:** 2025-05-27

**Authors:** Ester Gilart, Anna Bocchino, Patricia Gilart-Cantizano, Eva Manuela Cotobal-Calvo, Isabel Lepiani-Diaz, Daniel Román-Sánchez, José Luis Palazón-Fernández

**Affiliations:** 1Department of Nursing and Physiotherapy, University of Cádiz, 11009 Cádiz, Spain; ester.gilart@uca.es; 2Nursing Faculty “Salus Infirmorum”, University of Cádiz, Calle Ancha 29, 11001 Cádiz, Spain; evamanuela.cotobal@ca.uca.es (E.M.C.-C.); isabel.lepiani@ca.uca.es (I.L.-D.); daniel.roman@ca.uca.es (D.R.-S.); jluis.palazon@uca.es (J.L.P.-F.); 3Hospital la Linea de la Concepción, 11300 Cadiz, Spain; patricia.gilart@gmail.com

**Keywords:** expert panel, nursing professionals, artificial intelligence, diagnostic process

## Abstract

**Background/Objectives**: Nursing diagnosis is a complex process that requires clinical judgment, time, and resources and whose implementation is hindered by factors such as workload, lack of time, and resistance to computerized systems. This study aimed to compare the quality and efficiency of care plans generated by nursing professionals versus those produced by an artificial intelligence (AI) model, using the NANDA, NOC, and NIC taxonomies as criteria. **Methods**: An observational study was carried out with three simulated clinical cases. Thirty experts, fifty-four nursing professionals, and the ChatGPT model (GPT-4) were included. The experts established the referral plans using the Delphi technique. Responses were evaluated with a validated rubric (EADE-2) and analyzed using nonparametric tests. Professionals’ perceptions on the use of computer systems were also collected. **Results**: ChatGPT scored significantly higher on several dimensions (*p* < 0.001) and resolved all three cases in 35 s, compared to an average of 30 min for practitioners. Professionals expressed dissatisfaction with current diagnostic documentation systems. **Conclusions**: AI demonstrates high potential in optimizing the diagnostic process in nursing, although for its implementation human supervision, ethical aspects and improvements in current systems must be considered to achieve effective integration.

## 1. Introduction

Nursing diagnosis is a key component of individualized care planning and requires specialized clinical judgment [1]. This diagnosis is fundamental in the nursing care process, as it helps professionals to determine the care plan for their patients and to identify health needs. In addition, it allows for the planning and provision of individualized and effective care, prompting possible interventions for the patient [2,3,4].

It is a time, resource, and knowledge-intensive process that somewhat delays treatment and increases costs [5,6].

In clinical practice, the implementation of the nursing process plays a fundamental role in guaranteeing comprehensive, efficient care focused on the individual needs of the patient. In Andalusia, the DIRAYA system, a regional electronic health record platform, facilitates the integration of the nursing process by allowing for the documentation of standardized diagnoses, results, and interventions based on NANDA, NIC, and NOC taxonomies [7]. This system enables nurses to record patient data, formulate care plans, and update interventions in real time, enhancing the continuity and consistency of care [8]. However, factors such as clinical judgment, workload, care pressure, and the learning curve associated with its use influence its effectiveness [9].

Artificial intelligence (AI) has proven to be an effective tool in various healthcare fields, such as radiology, pathology, dermatology, and cardiology, where it has improved accuracy and speed in identifying health problems [10,11,12,13,14,15]. However, its application in nursing remains under development and requires further research and validation, especially due to the inherent complexity of nursing diagnoses and the variability in patient response [16].

The urgency to explore the potential of AI in nursing diagnoses is closely related to the current challenges of the healthcare system: increasing clinical complexity of patients, shortage of available professionals, and the need to provide rapid, individualized, and sustainable care. Added to this is the growing institutional pressure to incorporate technological solutions that optimize resources without compromising the ethical dimension and human quality of care [17].

Given this scenario, the need arises to evaluate the effectiveness of AI in comparison with care plans developed by nursing professionals. This study aims to address a critical gap in research by empirically comparing the diagnostic performance of an artificial intelligence system with that of human nursing professionals using standardized taxonomies (NANDA, NIC, and NOC). The objective is to evaluate the quality, efficiency, and applicability of AI-generated care plans in the context of increasing institutional pressure, staffing shortages, and the need for cost-effective and ethically sustainable solutions in nursing practice.

We hypothesized that ChatGPT would generate more complete and efficient care plans—according to evaluation criteria established by experts—than those devised by nursing professionals under conditions of care pressure.

## 2. Materials and Methods

### 2.1. Study Design

A descriptive observational study was conducted to compare the diagnostic performance and application of nursing interventions and outcomes by healthcare professionals and an artificial intelligence model (ChatGPT) in three simulated clinical cases.

### 2.2. Participants

The sample was composed of three groups:-Expert panel: 30 nurses with more than five years of clinical experience. They established the reference standard of correct responses (NANDA, NOC, NIC) using the Delphi technique. The consensus was reached over two rounds of anonymous and feedback-based evaluations, requiring ≥60% agreement to validate a given response.-Nursing professionals: 54 active nurses with a minimum of two years of experience, who completed the resolution of the cases using a structured questionnaire.-AI model (ChatGPT—OpenAI GPT-4): Provided answers to the three clinical cases acting as a nursing professional.

The sample size was calculated using the G-Power 3.1 program.

### 2.3. Instruments

Clinical cases: Three complex clinical cases, representative of common situations in the hospital setting, were selected based on previous peer-reviewed publications. Each case addressed different clinical dimensions: One focuses on hip arthroplasty in a patient with chronic comorbidities. Another focuses on a transient ischemic attack as a spontaneously resolving neurological event requiring health education, prevention, and anxiety management. The third focuses on non-ST-elevation acute coronary syndrome, a cardiovascular condition with relevant emotional and educational implications. The cases included basic contextual information (age, sex, reason for admission, clinical course) and were reviewed by the expert panel for relevance and diagnostic relevance. The selection of the three clinical cases responded to criteria of methodological feasibility and the need to maintain a manageable workload for the participants. Likewise, an attempt was made to ensure a balanced representation of different diagnostic domains of nursing practice—physical, emotional, and educational—in line with methodological precedents established in similar research [18,19,20,21].

Diagnoses (NANDA), interventions (NIC), and outcomes (NOC) were predefined by the expert panel through consensus.

Questionnaire:-Experts: An open-ended Google Form allowed experts to freely suggest appropriate NANDA, NIC, and NOC responses for each case.-Professionals and ChatGPT: A multiple-choice version of the questionnaire was provided, including the following:-Correct answers validated by the expert panel;-Distractor options generated by the research team;-Alternatives extracted from established databases such as DIRAYA.

In addition, a structured quantitative questionnaire was designed for the professionals, consisting of closed questions with a 5-point Likert-type scale (5: Strongly agree, 1: Strongly disagree). The instrument assessed perceptions and practices related to the use of nursing diagnoses and the NOC and NIC classifications in daily clinical practice.

In addition, multiple-choice questions were included to find out when professionals usually computerize these diagnoses and what software they use for this purpose.

Evaluation Rubric:

An adapted version of the validated “Nursing Diagnostic Accuracy Scale—Version 2 (EADE-2)” [22] was used to objectively and consistently assess responses. This rubric evaluates clinical competence in three domains: identification of nursing diagnoses (NANDA), selection of expected outcomes (NOC), and proposal of interventions (NIC). Each item was scored from 1 to 4 based on adequacy, coherence, and completeness.

The EADE-2 rubric has been previously validated to assess diagnostic competence in nursing and has demonstrated adequate internal consistency, with Cronbach’s alpha values above 0.80 in its principal components [22]. In the present study, inter-rater reliability was assessed using Cohen’s Kappa index, obtaining a value of κ = 0.72, indicating a substantial degree of inter-rater agreement. In cases of discrepancy, a third evaluator was used to reach consensus.

### 2.4. Procedure

Experts were recruited through direct contact with health service coordinators, followed by a snowball sampling strategy to identify other eligible professionals. For practicing professionals, purposive sampling was used, prioritizing diversity in clinical experience and areas of expertise. The number of experts was determined based on the recommendations of Polit and Beck [23,24] to ensure reliable estimates.

Prior to the main data collection phase, a pilot study was conducted with seven nursing professionals who met the same inclusion criteria as the target sample. The objective was to assess the clarity and usability of the clinical cases and the structured questionnaire. Participants were asked to identify possible ambiguities or difficulties in the wording of the items. Based on their comments, minor adjustments were made to improve comprehension. It was not necessary to modify the overall structure or content of the clinical cases.

Data were collected via Google Forms: open-format for experts and closed multiple-choice format for ChatGPT and professionals.

For ChatGPT (GPT-4), responses were generated using a standardized prompt to ensure consistency across the three clinical cases. Each prompt included the following instruction: “*Act as a nursing professional. Based on the clinical case provided below, review the multiple-choice options and select the most appropriate NANDA nursing diagnosis, NOC outcome, and NIC intervention. Please choose only one option for each category*”.

Brief definitions of the three taxonomies were provided at the beginning of the prompt as contextual guidance. The model selected its answers from the same multiple-choice options presented to nursing professionals. No additional training or adjustments were made to the model. Each clinical case was entered in a single interaction, and the responses obtained were recorded exactly as provided.

Two independent evaluators scored the responses using the rubric. In cases of disagreement, a third evaluator was consulted to reach a consensus and ensure reliability.

Inter-rater agreement was analyzed using Cohen’s Kappa index, with values above 0.60 considered acceptable.

This study was conducted in accordance with the 2013 Declaration of Helsinki (seventh revision, 64th meeting, Fortaleza) and the Organic Law 3/2018 of 5 December on the protection of personal data and guarantees of digital rights in Spain. Prior to data collection, the participants were informed about the objectives of the study, that their participation was completely voluntary (they could leave the study at any time if they did not feel comfortable for any reason), and that completion of the questionnaire implied they gave informed consent to participate. It was emphasized that the responses were anonymous and confidential in order to promote honesty. To ensure the privacy of the participants, only the research team had access to the data collected. The academic institutions granted approval for this study.

### 2.5. Data Analysis

The quantitative variable was expressed as the median and interquartile range. Categorical variables were described as frequencies and percentages. Frequencies and median scores for each participant across the three clinical cases and domains were calculated. Measures of central tendency (median) were estimated.

Due to the ordinal nature and lack of normality of the data, nonparametric tests such as Wilcoxon and Kruskal–Wallis were applied.

Cohen’s Kappa index was used to assess inter-rater agreement (threshold > 0.60 considered acceptable).

Data were analyzed using SPSS (version 26) for Windows (IBM Corp., Armonk, NY, USA). For all tests, *p*-values ≤ 0.05 were considered significant.

## 3. Results

### 3.1. Sociodemographic Characteristics of Expert Panel

The final sample was composed of a total of 30 experts, of whom 53.3% were nurses, 20% were specialist nurses, 10% were coordinators and/or supervisors, and 16.7% were teachers. Most of the participants were women (76.7%) with an average age of 43.20 (SD = 8.2) years. With regard to academic training, 50% had a diploma/degree in nursing, 20% had a specialty, and 30% undertook postgraduate studies (6.7% official master’s degree and 23.3% a doctorate). Regarding the area of work experience, most of the participants indicated that they were involved in clinical practice (83.3%). The average time of work experience was 16.27 (SD = 6.37) years. The above results are presented in detail in the following table (Table 1).

### 3.2. Sociodemographic Characteristics of Nursing Professionals

A total of 54 participants took part in the resolution of simulated clinical cases by nursing professionals. They were mostly women (75.9%), with an average age of 43 years. With respect to academic background, 44.4% of the participants were nurses with about 18.98 years of work experience. Their main characteristics can be observed in Table 2.

Analysis of the Likert-type items identified six main thematic clusters in nursing professionals’ perceptions of the NANDA, NOC, and NIC systems:Perceived time burden. More than half of the participants (51.8%) felt that using these classifications is too time-consuming, suggesting that documentation based on these systems represents a significant practical barrier.Perceived value. Standardized diagnoses were viewed positively by 51.8% as useful tools for clinical reasoning and care planning. However, 29.6% disagreed, reflecting a divided view of their actual applicability.Dissatisfaction with the system. Only 22.2% expressed satisfaction with the digital tools linked to these systems, compared to 38.9% who expressed dissatisfaction. This disparity suggests a gap between the theoretical potential of the classifications and their actual implementation.Limited practical application. Only 16.7% stated that they used these classifications during relays or in daily care work, which indicates the scarce transfer of theoretical knowledge to clinical practice.Perceived barriers. A total of 61.1% identified institutional or technical barriers that hinder the effective integration of these systems into their professional routine.Need for improvement. Finally, 70% of the participants expressed the need to update and modernize the documentation systems in order to facilitate their use and better adapt them to the real needs of the clinical setting.

The detailed distribution of responses is presented in Table 3.

In addition, the results indicate that 46.3% of the participants stated that they do not document diagnoses, NOC, or NIC. Some 40.7% do so during the shift, and a smaller percentage do so at the beginning (5.6%) or end (7.4%) of the shift.

The most mentioned program was Diraya, with 22 mentions. Other systems used include HCDM, GACELA, Casiopea, Hcis, and Drago, among others, although with significantly lower frequency. A total of 13 professionals indicated that they do not use any computer system to record diagnoses, NOC, and NIC, and others mentioned that in their current setting, no digital recording is implemented at all.

### 3.3. Delphi Technique for Expert Panel

Table 4 shows the results related to nursing diagnoses (NANDA), expected outcomes (NOC), and interventions (NIC) agreed by the Delphi technique in two rounds, considering only those with a level of agreement equal to or higher than 60%.

### 3.4. EADE-2 Score Rubric for Nursing Professionals and ChatGPT

#### 3.4.1. Nursing Professionals

Each participant was evaluated across three cases and three components (NANDA, NOC, NIC), each worth a maximum of 4 points.

#### 3.4.2. ChatGPT

Responses were generated by the AI model for the same clinical cases and assessed using the same rubric applied to professionals.

#### 3.4.3. Comparative Scores—Wilcoxon Test

In all comparisons (Table 5), statistically significant differences were found (*p* < 0.001), indicating differing response distributions between professionals and ChatGPT.

In the case-by-case analysis, there were differences in the degree of agreement between the ChatGPT responses and the expert panel consensus. In Case 1, the IA fully agreed with the NANDA diagnosis and had only slight discrepancies in the selection of NOC outcomes and NIC interventions. In Case 2, agreement was partial, with agreement on the NANDA diagnosis, but divergence on both the expected outcome (NOC) and the selected intervention (NIC). Case 3 showed the least agreement, especially for the NOC outcomes and the NIC interventions, where AI chose options poorly adjusted to the clinical context of the patient. In general, the model demonstrated greater terminological precision, a more complete response structure, and the absence of omissions, which translated into higher scores according to the criteria of the EADE-2 rubric.

#### 3.4.4. Response Times

Healthcare professionals required an average of 30 min per diagnosis, while ChatGPT completed the task for all three clinical cases in 35 s.

## 4. Discussion

From the analysis of the results obtained, the following findings stand out:-Although ChatGPT scored higher than nurses on the NANDA, NOC, and NIC components, these differences—while statistically significant—should be interpreted with caution. The practical implications depend on factors such as clinical complexity, practitioner experience, and care setting. The better performance of the AI model is largely due to its alignment with standardized classification systems and the terminological and structural consistency of its responses. In general, the model showed greater terminological accuracy, a more complete response structure, and the absence of omissions, which favored its scores according to the EADE-2 rubric criteria.

However, it is important to distinguish between diagnostic accuracy and contextual appropriateness. While ChatGPT showed a high correspondence with NANDA, NOC, and NIC taxonomies, its responses did not always reflect individualized decision making. Human professionals tend to adapt their reasoning based on patient priorities, psychosocial variables, and environmental factors, dimensions that AI, in its current form, is not yet able to replicate [25]. This reinforces the idea that artificial intelligence should be conceived as a complementary support tool, not as a substitute for clinical judgment.

-The lack of unanimous expert consensus highlights the complexity of the clinical cases analyzed. As already pointed out by different authors, it is difficult to establish diagnoses, outcomes, and interventions based solely on the reading of clinical cases, without taking into account the clinical history of each patient and/or the direct experience of the professional, which are key factors for a comprehensive and holistic assessment [26,27,28]. Indeed, among the possible current limitations of AI, recent studies highlight its inability to effectively manage real patient situations and emphasize the need for human supervision in nursing care planning [29,30,31].-Healthcare professionals are clearly ambivalent about the use of Diraya. In fact, the results highlight that a significant part of the participants consider that these procedures take up too much time in daily practice, and many of them do not apply them systematically in their unit or during shift reliefs. This trend could be explained by the fact that, although nurses recognize that these tools add value to clinical practice, there is a considerable level of dissatisfaction with the diagnostic systems available. Such dissatisfaction seems to be linked to the perception that these tools diminish the quality of interaction with patients, as noted in previous studies [32], undermining, in many cases, their clinical judgement and autonomy [33].-The majority agrees on the need to update the methodology for documenting nursing diagnoses, NOC, and NIC in order to optimize their use in clinical practice. This need has also been pointed out in previous research [34]. The observed dissatisfaction could be partly attributed to frequent incomplete documentation and a lack of relevant information in nursing records, which compromises the quality of care [35]. Factors contributing to these limitations include the fragmented nature of health systems, which makes effective integration of clinical information and standardization of documentation processes difficult [36].-Detailed analysis by case allowed us to identify variations in the performance of the AI model as a function of clinical complexity. In Case 1, which presented a clearer and more structured symptomatology, the AI model showed a high level of agreement with the expert panel, especially in the NANDA diagnosis. In contrast, in Case 3, which required greater contextual interpretation skills, the greatest discrepancies were observed, especially in NOC outcomes and NIC interventions. These findings suggest that AI performance is not homogeneous and may be conditioned by the clarity of clinical data and the need for inferential reasoning. Thus, although language models such as ChatGPT may be useful as decision support [37,38,39], their application should be evaluated with caution, particularly in settings where clinical judgment and personalization of care are imperative.-A noteworthy aspect of this study is the difference in the time required for the resolution of clinical cases between ChatGPT and the nursing professionals. This disparity evidences an advantage in terms of operational efficiency on the part of AI. However, such an advantage should be viewed with caution, as clinical decision making in nursing involves ethical, experiential, and contextual factors that exceed the current capabilities of automated systems. Therefore, although AI can streamline processes and reduce workloads, its use must be integrated within a framework that prioritizes human clinical judgment and professional oversight.-Despite the promising results obtained in the present study, it is essential to keep in mind a number of limitations that could be addressed in future research to strengthen the validity and applicability of the findings:-The use of a sample of experts in this study implies certain limitations in terms of the subjectivity and generalizability of the results. The subjective nature of expert opinions may introduce bias into the findings. Likewise, the generalizability of the results to international populations is limited, given that the questionnaire was developed taking into account the cultural characteristics of a single country (Spain).-The recruitment process of the participants, based on purposive and snowball sampling, may represent another limitation. This approach may have generated selection bias, favoring the inclusion of individuals with greater predisposition or familiarity with digital tools and standardized taxonomies. Consequently, the observed perceptions and diagnostic performance may not fully represent the diversity of the general nursing population.-An additional limitation of this study relates to the use of the multiple-choice format for both the nursing professionals and the ChatGPT model. While this format facilitated objective comparison between the two groups, it may have restricted the expression of more nuanced clinical reasoning or valid options that were not contemplated among the predefined responses. It is suggested that future research consider the use of mixed response formats to capture a wider range of diagnostic reasoning and allow for a more in-depth evaluation of AI-generated content.-Furthermore, although AI has shown greater technical accuracy, its performance could vary significantly depending on the diversity of the data as its accuracy could be affected in another cultural context, underlining the need for culturally adapted algorithms.

## 5. Conclusions

The integration of AI into nursing practice has significant potential to improve diagnostic accuracy, optimize clinical workflows, and anticipate patient needs. In addition, AI can help reduce errors, improve response times in emergency situations, and promote safer and more efficient care. This allows nurses to focus on more complex and value-added interventions that require clinical judgment, empathy, and specific and contextual care based on patients’ needs.

However, these benefits should be interpreted with caution. The effectiveness of AI depends largely on the quality and structure of the data it receives, as well as the design of prompts. Moreover, current systems still have significant limitations in adapting to the individual patient’s context and making value-based decisions. Therefore, AI should be conceived as a complementary tool, able to support—but not replace—nursing professionals.

To ensure correct implementation, however, it is essential to offer specific training to professionals and to encourage interdisciplinary collaboration in the development of these technologies.

Future efforts should be directed toward the design of hybrid systems that combine the technical accuracy of AI with the clinical intuition and ethical reasoning of experienced professionals. Future research should focus on conducting pilot studies in real-world clinical settings, using real patient cases, exploring longitudinal outcomes, and culturally adapting AI models to diverse healthcare environments to ensure safe, equitable, and context-sensitive implementation. Only through responsible, patient-centered, and ethically grounded integration can artificial intelligence truly contribute to improving the quality of nursing care.

## Figures and Tables

**Table 1 nursrep-15-00186-t001:** Sociodemographic characteristics of expert panel.

Variables	N	Mean (SD)	Percentage (%)
Sex			
Male	7		23.3
Female	23		76.7
Age		43.20 (8.2)	
Academic Qualification			
Degree in Nursing	2		6.7
University Diploma	13		43.3
Master’s Degree	2		6.7
Specialty	6		20.0
Doctorate	7		23.3
Position in the Institution			
Nurse	16		53.3
Specialist Nurse	6		20.0
Coordinator and/or Supervisor	3		10.0
Teaching	5		16.7
Area of Professional Practice			
Teaching	5		16.7
Research	0		0.0
Management	0		0.0
Clinical Practice	25		83.3
Years of Work Experience		16.27 (6.37)	

**Table 2 nursrep-15-00186-t002:** Sociodemographic characteristics of nursing professionals.

Variables	N	Mean (SD)	Percentage (%)
Sex			
Male	13		24.1
Female	41		75.9
Age		43.0 (11.63)	
Years of Work Experience		18.98 (11.85)	
Academic Qualification			
General Nurse	24		44.4
Specialist Nurse	21		38.9
Master’s Degree	5		9.3
Doctorate	4		7.4
Unit/Service			
Inpatient Ward	8		14.8
ICU	3		5.6
Operating Room	5		9.3
Primary Care	8		14.8
Emergency and Pre-Hospital Services	12		22.2
Hospital Emergency Department	3		5.6
Gynecology and Obstetrics Ward	3		5.6
Critical Care and Emergency Devices	1		1.9
Other	11		20.4
Position in the Institution			
Nurse	20		37.04
Nurse—Critical Care Area	2		3.7
Nurse—Operating Room Area	4		7.41
Nurse—Mental Health Area	2		3.7
Nurse—Emergency and Pre-Hospital Area	6		11.11
Nurse—Hospital Emergency Area	2		3.7
Specialist—Family and Community Nursing	1		1.85
Specialist—Obstetric–Gynecologic Nursing	12		22.22
Specialist—Pediatric Nursing	1		1.85
Specialist—Mental Health Nursing	1		1.85
Other	3		5.56
Years in Current Position		10.0 (9.75)	

**Table 3 nursrep-15-00186-t003:** Perception and utility of informatic system for Nanda, Noc, and Nic.

Item	Strongly Agree	Agree	Neutral	Disagree	Strongly Disagree
Do nursing diagnoses and NOC/NIC classifications take a lot of time in daily practice?	12	16	17	3	6
Do nursing diagnoses add value to clinical practice?	9	19	11	4	11
Are you satisfied with the current diagnostic tools available in your clinical practice?	2	10	21	9	12
Are NOC/NIC classifications applied practically in your workplace?	5	9	12	11	17
Is the time spent documenting diagnoses and care justified by patient benefits?	5	8	12	14	15
Are nursing diagnoses applied consistently within your team/unit?	3	8	11	11	21
Do you include nursing diagnoses and NOC/NIC classifications during shift handovers?	2	9	10	10	23
Do you follow your colleagues’ nursing diagnoses and NOC/NIC classifications at the beginning of your shift?	2	9	13	8	22
Could you work effectively without following nursing diagnoses and NOC/NIC classifications?	23	10	11	7	3
Do professionals review nursing diagnoses, NOC indicators, and NIC interventions at the start of their shift?	1	5	7	14	27
Do you use NOC indicators to assess the patient’s, caregiver’s, family’s, or community’s status before and after interventions?	4	10	7	9	24
Are there barriers in your workplace that hinder the implementation of nursing diagnoses and NOC/NIC classifications?	25	8	11	6	4
Do you think documentation methodology for nursing diagnoses, NOC, and NIC should be updated to optimize their use in clinical practice?	29	9	9	0	7

**Table 4 nursrep-15-00186-t004:** Nursing diagnoses (NANDA), expected outcomes (NOC), and interventions (NIC) reached by expert consensus using the Delphi technique (≥60% agreement).

Case	NANDA Diagnosis	% Agreement NANDA	NOC Outcome	% Agreement NOC	NIC Intervention	% Agreement NIC
Case 1	Impaired physical mobility	76.7%	Mobility	74.5%	Exercise therapy	83.3%
Case 1	Anxiety	63.3%	Anxiety self-control	63.9%	Anxiety reduction	64.1%
Case 1	Risk of impaired skin integrity	60%	Tissue integrity	62.7%	Pressure injury prevention	61.1%
Case 2	Deficient knowledge	86.7%	Knowledge: therapeutic regimen	71.3%	Teaching: disease process	61.8%
Case 2	Anxiety	86.7%	Anxiety self-control	66.6%	Anxiety reduction	81.5%
Case 2	Impaired physical mobility	66.3%	Mobility	65.3%	Exercise therapy	60.3%
Case 3	Anxiety	73.3%	Anxiety self-control	61.8%	Anxiety reduction/Sleep enhancement	84.3%
Case 3	Deficient knowledge	68.3%	Knowledge: therapeutic regimen	61.5%	Teaching: disease process	61.3%
Case 3	Impaired physical mobility	60%	Mobility	63.3%	Exercise therapy	63.3%

**Table 5 nursrep-15-00186-t005:** Comparative performance between nursing professionals and ChatGPT in NANDA, NOC, and NIC scoring across clinical cases (Wilcoxon test results).

Case	Category	Median Professionals	ChatGPT	Z Wilcoxon	*p*-Value	Effect Size (r)
CC1	NANDA	2	4	−6.405	0.000	−0.87
CC1	NOC	1	2	−5.657	0.000	−0.77
CC1	NIC	1	2	−7.280	0.000	−0.99
CC2	NANDA	3	3	−4.113	0.000	−0.56
CC2	NOC	2	1	5.745	0.000	0.78
CC2	NIC	2	2	−4.243	0.000	−0.58
CC3	NANDA	1	3	−6.707	0.000	−0.91
CC3	NOC	1	2	−5.831	0.000	−0.79
CC3	NIC	2	2	−4.690	0.000	−0.64

## Data Availability

The data presented in this study are available on request from the corresponding author. The data are not publicly available due to privacy restrictions.

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
