# Peer review of "The Integration of AI into the Nursing Process: A Comparative Analysis of NANDA, NOC, and NIC-Based Care Plans"

_nursrep, 2025, doi:10.3390/nursrep15060186_

Round 1

Reviewer 1 Report

Comments and Suggestions for Authors

The article addresses a topic relevant to nursing knowledge and practice, particularly with regard to the efficiency of nursing care, through objective indicators based on NANDA, NIC and NOC as evaluation criteria.

The authors point out some limitations and possibilities for improvement. It is structured according to the rules recommended by the Journal and presents coherence between the objective of the study, the discussion and the conclusion. A consultation was made of the bibliography pertinent to the reflections carried out, most of which were less than five years old.

Author Response

Dear Reviewer,

Thank you very much for your time and for the insightful comments and suggestions provided to improve our manuscript. We greatly appreciate your careful review and the opportunity to address your feedback.

Please find attached our detailed responses to each of your comments. We hope the changes we have made are satisfactory and have contributed to strengthening the manuscript.

Best regards,
Dr. Anna Bocchino

Reviewer 2 Report

Comments and Suggestions for Authors

Dear authors,

Congratulations on choosing such a timely and important topic, as well as on the overall quality of your manuscript. To help strengthen the work, I would like to offer some suggestions for improvement

The article addresses a highly relevant and timely topic, exploring how artificial intelligence (AI), specifically the ChatGPT model, can support the nursing diagnostic process by comparing its performance with that of human nursing professionals using standardized taxonomies. The manuscript is well-organized and engaging, but several aspects could be improved to increase its scientific rigor, clarity, and practical relevance.

The introduction provides a solid overview of the context, explaining the importance of the NANDA, NOC, and NIC taxonomies and describing the role of electronic systems such as DIRAYA in supporting nursing care documentation. It also appropriately situates the discussion within the broader landscape of AI applications in healthcare. However, the problem statement could be framed more sharply: why is it necessary or urgent to compare AI and human performance in this domain at this particular moment? Are there specific gaps in the literature, institutional pressures, cost-effectiveness concerns, or evolving ethical debates that make this study especially relevant now? Clarifying this would strengthen the study’s rationale. Additionally, the introduction sometimes relies on long chains of references that may feel overwhelming; synthesizing key points more concisely would improve readability. Importantly, the authors should explicitly highlight the research gap that this study seeks to address.

The materials and methods section is one of the article’s strengths, as it provides solid details about the participants, instruments (such as the Delphi process and the EADE-2 scale), and statistical analyses. However, several improvements are needed. It is important to clarify the recruitment process: how were participants selected, and was there any risk of selection bias? Furthermore, the clinical cases used in the study need to be described in more depth — simply stating that they were drawn from the literature is insufficient to ensure the study can be replicated by others. Equally crucial is providing a detailed explanation of how the AI model was prompted: what exact inputs were given to ChatGPT, were there adjustments or refinements, and how were outputs selected or filtered? Given that the performance of large language models depends heavily on prompt engineering, transparent reporting of these procedures is essential to validate the results. Finally, the study would be strengthened by explaining why only three clinical cases were used — was this a logistical constraint, or was there a scientific rationale behind this decision?

The results section presents clear findings, using appropriate nonparametric statistical methods given the ordinal nature of the data. Including the professionals’ perceptions of the electronic systems was a valuable addition that enriches the understanding of the practical barriers present in current practice. However, the interpretation of the statistical differences could be framed more cautiously: although the AI achieved higher scores in some areas, do these differences translate into clinically meaningful improvements, or are they mainly statistically significant but practically modest? Incorporating visual elements, such as graphs or tables, would also greatly enhance the clarity of the results, particularly when summarizing the professionals’ opinions and the distribution of responses across various items.

The discussion appropriately addresses the study’s main limitations and places the results within the broader body of literature, acknowledging that while AI shows promise in optimizing diagnostic processes, it cannot replace human supervision or account for all contextual and ethical factors. However, the discussion could be deepened in several ways. Beyond stating that AI offers optimization, the authors should critically reflect on the potential risks, such as the dehumanization of care, overreliance on technology, algorithmic biases, and the potential impact on nurses’ professional autonomy and identity. These are essential considerations in any conversation about integrating AI into healthcare. Furthermore, the discussion could benefit from offering more concrete recommendations for practical implementation, such as proposing pilot tests in real clinical settings, developing tailored training for nursing teams, or designing culturally sensitive AI algorithms.

The conclusions are aligned with the presented findings and effectively summarize the study’s contributions. However, they would benefit from adopting a more cautious tone, avoiding overgeneralization, and explicitly acknowledging the exploratory nature of the study, which is based on a limited sample and simulated cases rather than real-world testing or multicenter trials. The article would also be strengthened by including concrete recommendations for future research, such as expanding the sample size, incorporating real patient cases, conducting longitudinal assessments, or examining patient-centered outcomes.

Author Response

Dear Reviewer,

Thank you very much for your time and for the insightful comments and suggestions provided to improve our manuscript. We greatly appreciate your careful review and the opportunity to address your feedback.

Please find attached our detailed responses to each of your comments. We hope the changes we have made are satisfactory and have contributed to strengthening the manuscript.

Responses to Reviewer 2's Comments

Dear authors,

Congratulations on choosing such a timely and important topic, as well as on the overall quality of your manuscript. To help strengthen the work, I would like to offer some suggestions for improvement

The article addresses a highly relevant and timely topic, exploring how artificial intelligence (AI), specifically the ChatGPT model, can support the nursing diagnostic process by comparing its performance with that of human nursing professionals using standardized taxonomies. The manuscript is well-organized and engaging, but several aspects could be improved to increase its scientific rigor, clarity, and practical relevance.

We would like to sincerely thank the reviewer for his detailed and constructive comments. We appreciate the acknowledgment of the interest and organization of the manuscript, and we have carefully considered each suggestion in order to improve the scientific rigor, clarity, and applicability of our study. Below, we respond point by point and note the changes made in the revised version of the manuscript.

The introduction provides a solid overview of the context, explaining the importance of the NANDA, NOC, and NIC taxonomies and describing the role of electronic systems such as DIRAYA in supporting nursing care documentation. It also appropriately situates the discussion within the broader landscape of AI applications in healthcare. However, the problem statement could be framed more sharply: why is it necessary or urgent to compare AI and human performance in this domain at this particular moment? Are there specific gaps in the literature, institutional pressures, cost-effectiveness concerns, or evolving ethical debates that make this study especially relevant now? Clarifying this would strengthen the study’s rationale.

Dear reviewer, thank you very much for this suggestion. We have reworded the introduction to highlight more clearly the current need to compare AI performance with that of nursing professionals in the diagnostic process. Specifically, we highlight the paucity of empirical studies evaluating AI-generated care plans using standardized taxonomies such as NANDA, NIC, and NOC Institutional pressures, cost-effectiveness concerns, and emerging ethical debates are also mentioned that reinforce the relevance of this research in the current context (Page 2 lines 55-71).

Additionally, the introduction sometimes relies on long chains of references that may feel overwhelming; synthesizing key points more concisely would improve readability. Importantly, the authors should explicitly highlight the research gap that this study seeks to address.

We appreciate this observation. We have reduced the number of consecutive references of the introduction and synthesized the key concepts to facilitate a smoother reading without sacrificing academic rigor (Page 2 lines 49-60).

The materials and methods section is one of the article’s strengths, as it provides solid details about the participants, instruments (such as the Delphi process and the EADE-2 scale), and statistical analyses. However, several improvements are needed. It is important to clarify the recruitment process: how were participants selected, and was there any risk of selection bias?

We have expanded the Procedure section to describe the recruitment process, based on a combination of purposive sampling and snowball technique through service coordinators (Page 3 lines 141-145). We also acknowledge the possibility of selection bias, which is now discussed in the Limitations section (Page 12 lines 353-358 ).

Furthermore, the clinical cases used in the study need to be described in more depth — simply stating that they were drawn from the literature is insufficient to ensure the study can be replicated by others.

Thank you very much for this comment. We agree with your suggestion and therefore, we have expanded the description of the clinical cases, including context (Page 2, lines 94-95; Page 3 lines 96-107).

Equally crucial is providing a detailed explanation of how the AI model was prompted: what exact inputs were given to ChatGPT, were there adjustments or refinements, and how were outputs selected or filtered? Given that the performance of large language models depends heavily on prompt engineering, transparent reporting of these procedures is essential to validate the results.

Thank you very much for your suggestion. We have added a detailed description of the prompting protocol used with the ChatGPT model, including the input structure, the inclusion of taxonomic definitions (NANDA, NOC, NIC), and the presentation of the same multiple-choice options that were given to nursing professionals. We clarify that no specific training or fine-tuning of the model was performed.

Additionally, we have revised the previous statement regarding the use of open-ended responses. ChatGPT was presented with the same multiple-choice questionnaire as the professionals, in order to ensure symmetry in data collection and allow for a more direct and objective comparison (Page 4 lines 156-165 ).

Finally, the study would be strengthened by explaining why only three clinical cases were used — was this a logistical constraint, or was there a scientific rationale behind this decision?

We would like to sincerely thank for his detailed and constructive comments. We have included a methodological justification in the Instruments section: three cases were selected for reasons of feasibility and workload for the experts, in addition to ensuring diagnostic variety (physical, emotional and educational dimensions). This choice is also consistent with previous studies using clinical simulations (Page 2, lines 94,95; Page 3 lines 96-107).

The results section presents clear findings, using appropriate nonparametric statistical methods given the ordinal nature of the data. Including the professionals’ perceptions of the electronic systems was a valuable addition that enriches the understanding of the practical barriers present in current practice. However, the interpretation of the statistical differences could be framed more cautiously: although the AI achieved higher scores in some areas, do these differences translate into clinically meaningful improvements, or are they mainly statistically significant but practically modest?

We appreciate this observation. We have clarified the interpretation of the results, in the discussion section, differentiating between statistical significance and clinical relevance (Page 10 lines 293-297 ).

 Incorporating visual elements, such as graphs or tables, would also greatly enhance the clarity of the results, particularly when summarizing the professionals’ opinions and the distribution of responses across various items.

Thank you very much for your valuable suggestion. While we ultimately chose not to include an additional figure in order to comply with the journal's length limitations and to avoid redundancy or excessive extension of the manuscript, we have added a clarifying paragraph to improve the interpretability of the results.

Specifically, we included a case-by-case analysis that highlights the differences in the degree of agreement between the responses provided by ChatGPT and the expert panel consensus. This new section serves to illustrate more clearly the areas of stronger or weaker alignment between the AI-generated responses and expert judgment, as you proposed.

We hope this textual clarification sufficiently addresses your recommendation to enhance the clarity and analytical depth of the results section (Page 9 lines 263-272).

The discussion appropriately addresses the study’s main limitations and places the results within the broader body of literature, acknowledging that while AI shows promise in optimizing diagnostic processes, it cannot replace human supervision or account for all contextual and ethical factors. However, the discussion could be deepened in several ways. Beyond stating that AI offers optimization, the authors should critically reflect on the potential risks, such as the dehumanization of care, overreliance on technology, algorithmic biases, and the potential impact on nurses’ professional autonomy and identity. These are essential considerations in any conversation about integrating AI into healthcare. Furthermore, the discussion could benefit from offering more concrete recommendations for practical implementation, such as proposing pilot tests in real clinical settings, developing tailored training for nursing teams, or designing culturally sensitive AI algorithms.

Thank you for your suggestion, we have expanded the discussion section to include these ethical and professional risks. Issues such as loss of autonomy, algorithmic bias, and the need for ethical regulation are addressed. Current recommendations on the safe and responsible use of AI in healthcare are also cited.

The conclusions are aligned with the presented findings and effectively summarize the study’s contributions. However, they would benefit from adopting a more cautious tone, avoiding overgeneralization, and explicitly acknowledging the exploratory nature of the study, which is based on a limited sample and simulated cases rather than real-world testing or multicenter trials. The article would also be strengthened by including concrete recommendations for future research, such as expanding the sample size, incorporating real patient cases, conducting longitudinal assessments, or examining patient-centered outcomes.

Thank you for your comment. We have incorporated specific recommendations into the manuscript, such as conducting pilot studies in real clinical environments, developing targeted training programs in digital literacy for nursing professionals, and promoting the integration of multidisciplinary teams in the design of AI algorithms. Furthermore, the revised conclusions explicitly suggest future lines of research, including longitudinal studies, the use of real patient cases, and the cultural adaptation of AI models to different healthcare settings (Page 12 lines 371-380: page 13, lines 381-393).

Reviewer 3 Report

Comments and Suggestions for Authors

Abstract

  • Poor readability due to odd line breaks and syntax issues (e.g., "12clinical judgment", "15intelli-").
  • Phrasing is sometimes awkward: "whose implementation is affected by various organizational factors" is vague.
  • Overly packed: tries to include too much (Delphi method, rubric, perception survey).
  • Clean up formatting.
  • Shorten for clarity. Focus on 3 main takeaways: what was tested, what was found, and what it means.
  • Move minor details (like rubric name or survey software) to Methods.

Introduction

  • Overly descriptive in some parts (e.g., how DIRAYA works)—this could be condensed.
  • Repeats concepts (e.g., complexity of nursing diagnosis is mentioned multiple times).
  • Vague goal statement: “this study aims to analyze” lacks specificity until the last sentence.
  • Replace general justifications with a clearly articulated research gap, e.g., “No previous studies have empirically tested AI’s performance using NANDA/NOC/NIC classifications.”
  • Provide a clearer hypothesis or expectation.
  • Tighten redundancy and technical detail.

Materials and Methods

  • Unclear questionnaire design: Why use multiple-choice for nurses and open-ended for experts/ChatGPT? This creates response modality bias.
  • No mention of case validation or pilot testing.
  • The role of the ChatGPT prompting protocol is underexplained. What input was provided? Was it consistent? Were system-level instructions included?
  • Clarify the ChatGPT setup: e.g., prompt format, whether it was primed with taxonomies or definitions.
  • Justify the asymmetry in data collection (experts: open; professionals/AI: multiple choice).
  • Add rationale for selecting three cases only.

Results

  • Redundant tables and figures: Some sociodemographic data could be summarized.
  • EADE-2 scoring results are sparse: No discussion of internal consistency or score reliability.
  • Some important contrasts (e.g., time vs. accuracy trade-off) are only briefly mentioned.
  • Table 3 (perceptions) is very large and repetitive; needs a more analytical summary.
  • Include effect sizes in Wilcoxon tests to complement p-values.
  • Summarize perception results with thematic clusters (e.g., "Perceived time burden" or "System dissatisfaction").
  • Add brief qualitative quotes or comment summaries if available.

Discussion

  • Redundant with the Introduction: Repeats that nursing diagnosis is time-consuming.
  • Doesn't explore why AI outperformed professionals. Was it due to completeness? Alignment with taxonomy? Efficiency?
  • Overgeneralizes about AI benefit without dissecting specific cases where AI was stronger or weaker.
  • Analyze how ChatGPT achieved higher scores—what patterns or content types were better?
  • Distinguish diagnostic precision vs. contextual accuracy.
  • Emphasize the complementary role of AI, rather than implying it outperforms humans across the board.

Conclusions

  • Overstates AI’s effectiveness without discussing data dependency, prompt tuning, or clinical context limits.
  • No mention of training needs for nurses to interact meaningfully with AI systems.
  • Clarify that AI is a support tool, not a replacement.
  • Recommend future hybrid systems that combine AI accuracy with clinician intuition.

Ethics, Author Contributions, and Disclosure

  • Ethics statement could be misinterpreted—states that review was unnecessary but then implies data was gathered from human participants.
  • Clarify why no ethics board review was needed, but reiterate voluntary consent and anonymization.
  • Consider elaborating the AI Use Declaration (e.g., clarify ChatGPT was part of the study, not the writing process).

Author Response

Dear Reviewer,

Thank you very much for your time and for the insightful comments and suggestions provided to improve our manuscript. We greatly appreciate your careful review and the opportunity to address your feedback.

Please find attached our detailed responses to each of your comments. We hope the changes we have made are satisfactory and have contributed to strengthening the manuscript.

Responses to Reviewer 3's Comments

We would like to sincerely thank the reviewer for his detailed and constructive comments. We have carefully considered each suggestion in order to improve the scientific rigor, clarity, and applicability of our study. Below, we respond point by point and note the changes made in the revised version of the manuscript.

Abstract

  • Poor readability due to odd line breaks and syntax issues (e.g., "12clinical judgment", "15intelli-").
  • Phrasing is sometimes awkward: "whose implementation is affected by various organizational factors" is vague.
  • Overly packed: tries to include too much (Delphi method, rubric, perception survey).
  • Clean up formatting.
  • Shorten for clarity. Focus on 3 main takeaways: what was tested, what was found, and what it means.
  • Move minor details (like rubric name or survey software) to Methods.

Dear reviewer, thanks for this comment. As a first step, we have thoroughly revised the abstract to enhance readability, eliminate unnecessary technical details, and emphasize the three key elements of the study: what was tested, what was found, and what it means (Page 1 lines 12-28).

Introduction

  • Overly descriptive in some parts (e.g., how DIRAYA works)—this could be condensed.
  • Repeats concepts (e.g., complexity of nursing diagnosis is mentioned multiple times).

Response:

Dear Review, we appreciate the feedback regarding the structure and focus of the Introduction section. In response, we have substantially restructured and condensed the Introduction to improve clarity, reduce redundancy, and eliminate unnecessary technical descriptions (Page 1, line 42; Page 2 lines 43-54).

  • Vague goal statement: “this study aims to analyze” lacks specificity until the last sentence.

Response:

Dear reviewer, thanks for this comment.  We have now revised it to explicitly state the objective of the study at the end of the Introduction:

This study aims to address a critical gap in research by empirically comparing the diagnostic performance of an artificial intelligence system with that of human nursing professionals using standardized taxonomies (NANDA, NOC and NIC). The objective is to evaluate the quality, efficiency, and applicability of AI-generated care plans in a context of increasing institutional pressure, staffing shortages, and need for cost-effective and ethically sustainable solutions in nursing practice.

(Page 2 lines 55-68)

  • Replace general justifications with a clearly articulated research gap, e.g., “No previous studies have empirically tested AI’s performance using NANDA/NOC/NIC classifications.”

Response

We have replaced general justifications with a clear statement of the research gap, emphasizing that, to our knowledge, no previous studies have empirically evaluated AI performance using NANDA/NOC/NIC classifications (Page 2 lines 55-68).

  • Provide a clearer hypothesis or expectation.

Response:

Thank you for this observation. We have now included a specific and testable hypothesis at the end of the Introduction:

We hypothesized that ChatGPT would generate more complete and efficient care plans, according to evaluation criteria established by experts, than those elaborated by nursing professionals under conditions of care pressure (Page 2 lines 69-71).

  • Tighten redundancy and technical detail.

Response:

We appreciate this observation. In the revised version of the manuscript, we have eliminated unnecessary repetitions, especially those related to the complexity of the nursing diagnosis, which was mentioned on several occasions. We have also reduced the level of technical detail in the Introduction section, particularly in the description of the DIRAYA system and the initial assessment process. These explanations have been synthesized into a more concise paragraph, while maintaining the relevant references to support the information presented (Page 2 lines 43-48).

Materials and Methods

  • Unclear questionnaire design: Why use multiple-choice for nurses and open-ended for experts/ChatGPT? This creates response modality bias.

Response:

Thank you for this observation, which has led us to revise and improve the methodological consistency of the study. In the revised version, we clarified that both nursing professionals and the ChatGPT model responded to the same structured multiple-choice questionnaire, with differentiated blocks for NANDA diagnoses, NOC outcomes, and NIC interventions. Initially, the design included open-ended responses for ChatGPT, but after reflecting on its methodological impact, we decided to unify the format to ensure better comparability between groups and reduce possible biases derived from the type of response.The experts, however, continued to use an open format, given that their objective was not direct comparison, but rather prior validation of the items using the Delphi technique to establish the reference standard. This difference is now more clearly explained in the Materials and Methods section (Page 4 lines 156-164).

  • No mention of case validation or pilot testing.

Response:

Dear reviewer, we thank you for this comment. It is true that a small pilot study was conducted with 7 nurses who met the same inclusion criteria as the final sample. The participants reviewed the clinical cases and the multiple-choice questionnaire to assess the clarity, relevance and ease of understanding of the items. Based on their input, minor adjustments were made to the wording of some response options to improve their comprehensibility. This information has been incorporated in the procedure section (Page 4 lines 147-153).

  • The role of the ChatGPT prompting protocol is underexplained. What input was provided? Was it consistent? Were system-level instructions included?

Response:

Thank you for this comment, which has allowed us to improve the methodological transparency of the study. In the revised version of the manuscript, we have included a detailed description of the protocol for using ChatGPT in the Materials and Methods section. All clinical cases were entered using a standardized prompt, with consistent instructions for all three cases. The prompt included:

  1. A general instruction for the model to act as a nursing professional.
  2. The full text of the clinical case.
  3. Brief definitions of the NANDA, NOC and NIC taxonomies.
  4. A final instruction to select the most appropriate option from each block (diagnosis, outcome, intervention), from a closed list of options identical to that used by the professionals.

No specific tuning system or system-level instructions were used. Each case was entered in a single interaction, with no subsequent modification of the responses generated (page 4, lines 156-165).

  • Clarify the ChatGPT setup: e.g., prompt format, whether it was primed with taxonomies or definitions.

Response

Thank you for the comment. In the revised version of the manuscript we have clarified the configuration used with the ChatGPT model (Page 4 lines 154-165).

Justify the asymmetry in data collection (experts: open; professionals/AI: multiple choice).

Response:

Thank you for your comment. In the revised version of the manuscript, we have incorporated a  justification for this methodological decision.

The asymmetry in the response format responds to the different roles of the participating groups:

The expert group used an open format because its objective was not direct comparison, but to generate and validate the diagnoses, results and reference interventions for each case, using the Delphi technique.

On the other hand, both the nursing professionals and the ChatGPT model completed a structured multiple-choice questionnaire, designed from the answers agreed upon by the experts. This strategy allowed an objective and controlled comparison between both groups evaluated (page 12, lines 359-365).

Add rationale for selecting three cases only.

Thank you for your comment. We have incorporated a justification for the selection of the three clinical cases used in the study (Page 2 lines 94-95; Page 3, lines 96-107). The decision to work with three cases was based on criteria of methodological feasibility. We sought to include representative clinical cases, with varying degrees of complexity, that would allow us to evaluate the performance of the AI model and of the professionals without overloading the participation time. In addition, limiting the number of cases favored a detailed evaluation by the panel of experts using the Delphi technique, ensuring the quality of the reference standard. In future studies, it is recommended that the number of cases be expanded to explore the generalizability of the results.

Results

  • Redundant tables and figures: Some sociodemographic data could be summarized.

Response

Dear reviewer, thank you very much for your comment. While we have considered the possibility of summarizing the sociodemographic data further, we believe it is important to retain the current format of the table in order to enhance clarity and facilitate the interpretation of differences between the two distinct participant groups: nursing professionals and expert reviewers. Presenting the data side-by-side in a single comparative table (Table 1) allows readers to clearly distinguish the characteristics of both samples, which is relevant for understanding their respective roles and contributions within the study.

  • EADE-2 scoring results are sparse: No discussion of internal consistency or score reliability.

Response

Thank you for your valuable comment. We have now added information regarding the psychometric properties of the EADE-2 rubric. Specifically, we included a reference to the original validation study, which reported Cronbach’s alpha values above 0.80 for internal consistency. In addition, we reported the inter-rater reliability obtained in our study (κ = 0.72), indicating substantial agreement between the evaluators. This information has been added to the "Instruments" section (2.3) of the manuscript. (Page 3 lines 132-137).

  • Some important contrasts (e.g., time vs. accuracy trade-off) are only briefly mentioned.

Response

Thank you for your valuable comment. We have now added information regarding this aspect in the discussion.

  • Table 3 (perceptions) is very large and repetitive; needs a more analytical summary.

Response

Thank you for your comment. We agree that an analytical summary improves the clarity of the results, and therefore, thanks to your suggestion, we have added one before table 3 (Page 7 lines 213-233). However, we consider it important to keep the table complete, as each item reflects a specific dimension of professional perception (such as time burden, satisfaction, practical use or documentation habits). These categories allow us to capture important nuances about the actual use of standardized diagnostic systems in clinical settings, a rather contradictory argument in the Spanish context, especially in Andalusia.

  • Include effect sizes in Wilcoxon tests to complement p-values.

Response

Dear reviewer, thank you very much for the suggestion. We have already incorporated this value in table 5.

  • Summarize perception results with thematic clusters (e.g., "Perceived time burden" or "System dissatisfaction").

Response

Thank you for your suggestion. We have reorganized the perception results into six thematic clusters to improve clarity and synthesis. These clusters are now explicitly described in the Results section (3.4), with references to detailed item distributions provided in Table 3 (Page 7 lines 213-233).

  • Add brief qualitative quotes or comment summaries if available.

Response

Thank you for your suggestion. Qualitative data from participant comments have indeed been collected and are currently being analyzed and presented in a separate manuscript, which is under review for publication. Since the main objective of the present article was to compare the accuracy and diagnostic selection between nursing professionals and an AI model using structured quantitative data, we considered it more appropriate to reserve the qualitative findings for a complementary, dedicated analysis. We appreciate your understanding.

Discussion

  • Redundant with the Introduction: Repeats that nursing diagnosis is time-consuming.

Response

Thank you for your comment. We have revised the Discussion section to eliminate repetitive statements that were already mentioned in the Introduction. The time burden associated with the use of standardized diagnostic systems is retained only where it contributes relevant interpretive analysis to the results.

  • Doesn't explore why AI outperformed professionals. Was it due to completeness? Alignment with taxonomy? Efficiency?

Response

  • Thank you for this comment. We have incorporated in the Discussion section a new paragraph that clearly differentiates between diagnostic accuracy (defined as formal alignment with NANDA-I, NOC and NIC taxonomies) and contextual appropriateness (related to individualized clinical adaptation and professional judgment). We believe that this distinction is key to interpreting the results and establishing a complementary and realistic use of artificial intelligence in the care setting (Page 10 lines 385-392).
  • Overgeneralizes about AI benefit without dissecting specific cases where AI was stronger or weaker.
  • Analyze how ChatGPT achieved higher scores—what patterns or content types were better?

Response

We very much appreciate these observations. To address it, we have added a detailed case-by-case analysis in the Results section, describing the variations in agreement between ChatGPT and expert panel responses in the three clinical cases (Page 9 lines 263-272).

This analysis has also been expanded in the Discussion section, where we reflect on the contexts in which AI showed higher or lower performance. These additions allow us to offer a more nuanced interpretation of the capabilities and limitations of the model.

This analysis has also been expanded in the Discussion section offering an interpretation of the results obtained (Page 11 lines 326-336).

  • Distinguish diagnostic precision vs. contextual accuracy.
  • Emphasize the complementary role of AI, rather than implying it outperforms humans across the board.

Response:

We are grateful for these comments, which have been key to improving the interpretive approach of the manuscript. In the revised version, an explicit distinction between diagnostic accuracy (technical alignment with NANDA, NOC and NIC taxonomies) and contextual appropriateness (ability to adapt to the individual patient's context) has been incorporated. This differentiation is now clearly developed in the first paragraphs of the Discussion.

In addition, the overall discourse has been reworded to avoid any message suggesting a general superiority of AI over human professionals. Instead, it is emphasized that the role of artificial intelligence should be understood as complementary, especially useful in structured or technical tasks, but always under professional supervision and within a patient-centered framework.

These improvements are reflected in paragraphs 1 and 2 of the Discussion section.

Conclusions

  • Overstates AI’s effectiveness without discussing data dependency, prompt tuning, or clinical context limits.
  • No mention of training needs for nurses to interact meaningfully with AI systems.
  • Clarify that AI is a support tool, not a replacement.
  • Recommend future hybrid systems that combine AI accuracy with clinician intuition.

Response

We are grateful for your comments, which have been instrumental in improving the balance and depth of the Conclusions section. In the revised version, this section has been modified to incorporate all the suggestions made.

Ethics, Author Contributions, and Disclosure

  • Ethics statement could be misinterpreted—states that review was unnecessary but then implies data was gathered from human participants.
  • Clarify why no ethics board review was needed, but reiterate voluntary consent and anonymization.

Response

Dear Reviewer, thank you for your clarification. We have revised the paragraphs you mentioned to clarify any potential misunderstandings.

  • Consider elaborating the AI Use Declaration (e.g., clarify ChatGPT was part of the study, not the writing process).

Response

 Thank you for this observation. In the revised version of the manuscript, the Statement on the use of artificial intelligence has been updated to better specify your suggestions.

Round 2

Reviewer 2 Report

Comments and Suggestions for Authors

Dear,

Accept in present form.

Reviewer 3 Report

Comments and Suggestions for Authors

The authors addressed all the comments. I have no more comments.